# Over the Limits of Traditional Sampling: Advantages and Issues of AICs for Measurement Instrumentation

**DOI:** 10.3390/s23020861

**Published:** 2023-01-11

**Authors:** Grazia Iadarola, Pasquale Daponte, Luca De Vito, Sergio Rapuano

**Affiliations:** 1Department of Information Engineering, Polytechnic University of Marche, 60131 Ancona, Italy; 2Department of Engineering, University of Sannio, 82100 Benevento, Italy

**Keywords:** Analog-to-Information Converter, measurement instrumentation, metrological characterization, wideband acquisition, sparse signals, Compressive Sampling, sub-Nyquist sampling, data compression

## Abstract

Data acquisition systems have shown the need of wideband spectrum monitoring for many years. This paper describes and discusses a recently proposed architecture aimed at acquiring efficiently wideband signals, named the Analog-to-Information Converter (AIC). AIC framework and working principle implementing the sub-Nyquist sampling are analyzed in general terms. Attention is specifically focused on the idea of exploiting the condition of the signals that, despite their large bandwidth, have a small information content in the frequency domain. However, as clarified in the paper, employing AICs in measurement instrumentation necessarily entails their characterization, through the analysis of their building blocks and the corresponding non-idealities, in order to improve the signal reconstruction.

## 1. Introduction

Through the years, spectrum monitoring has become increasingly important and measurement instrumentation has an essential role in this regard. In fact, spectrum sharing among different users that take advantage of the same frequency range involves continuous measurements to avoid interfering emissions or congestions for real-time transmissions. For example, in civil matters, not only is spectrum monitoring useful to institutions that allocate spectrum portions, but also to ensure the security of critical infrastructures such as airports or embassies. Moving on to the military field, spectrum monitoring is quite sought-after in electronic warfare, for detecting and analyzing an immediate threat, or for long-term plans. The frequency spectrum adopted by recent communication systems ranges from dozens of MHz to hundreds of GHz and it is gradually destined to expand due to continuous innovations on Very Large Scale Integration technologies. Thus, measurement instrumentation should guarantee a proper monitoring with good wideband capabilities. Unfortunately, traditional data acquisition systems are not appropriate to support the growing demand for wideband spectrum sensing.

Nowadays, *Compressive Sampling* (CS) has paved the way for a more efficient monitoring of the spectrum than in the past. By means of CS, research has demonstrated that, also in some conditions where equivalent time sampling cannot be adopted, the sampling frequency can be pushed below the Nyquist rate, which represents, instead, the limit for traditional Analog-to-Digital Converter (ADC). The CS paradigm works correctly when signals to be sampled are compressible in the frequency domain, i.e., when, after Fourier transform, they can be represented by a few relevant components. Anyway, compressibility in the frequency domain is a condition recurring in a considerable part of signals. CS has further inspired a new architecture for analog signal acquisition: the *Analog-to-Information Converter* (AIC) [1,2]. AICs are innovative devices, intended to acquire the information content of given signals of interest. They are useful in so many fields, especially for radio spectrum monitoring, since they are able to implement in practice the sub-Nyquist sampling, overcoming the existing limits of data acquisition and digital communications. However, the use of AICs entails the need for their characterization, through the analysis of building blocks and corresponding non-idealities [3], in order to assess their capability in signal reconstruction.

In light of the foregoing, the contribution of this article is intended to provide a perspective on AICs, showing their advantages, and the technical issues related to their use in measurement instrumentation. In detail, Section 2 illustrates the background and the motivation behind the conception of AICs. Section 3 describes the working principle of a generic AIC, defining the measurands. Section 4 focuses on the metrological characterization in AICs, which is aimed to guarantee a good signal reconstruction. Finally, Section 5 draws the concluding remarks.

## 2. Wideband Acquisition from Nyquist-Shannon to CS Paradigm

The foundations for modern digital communications have been laid as far back as in the first half of the twentieth century on the Nyquist-Shannon theorem, from which the sampling theory originated. The Nyquist-Shannon theorem indicates that, for a proper reconstruction of a band-limited analog signal, the sampling frequency $f_{s}$ should be at least equal to twice the signal bandwidth *B*, i.e., the Nyquist rate $f_{Nyq}$ of the signal:(1)$$f_{s}\geq f_{Nyq}=2B.$$
If such requirement on the sampling frequency is not satisfied, the aliases overlap with the signal and the exact reconstruction is unattainable. The sampling theory represents a landmark in signal processing since it relates any natural signal to an artificial stream of integer numbers that is easier to manage and transmit. By rigorously obeying the Nyquist-Shannon rule, ADCs generally provide input signal samples that are uniformly spaced. All hardware technology implementing digital signal processing essentially revolves around ADCs, since the interaction with the analog communication world is uniquely entrusted to them.

Wideband communications open a gap with traditional sampling techniques not only because of the need for a high sampling frequency. In fact, large amounts of samples are required to represent wideband signals. There are several consequences produced if a big quantity of data is involved, such as trouble to find relevant information, burden to process signals with many coefficients and high power consumption, large resources for storage and transmission. Besides, technological limits impose that a high sampling frequency is necessarily counterbalanced by a low resolution [4]. Thus, sampling at a high frequency is not convenient from different points of view.

Henceforth, let a specific condition concerning signals be considered: the sparsity in the frequency domain. Sparsity in the frequency domain represents the condition that a signal x(t) carries an information content occupying only a small portion of the signal bandwidth. Therefore, the information content can be acquired at a rate much lower than the bandwidth of the signal, that is, at the *information rate*.

In general, the set of the *N* samples ${\left\{x\left(n\right)\right\}}_{n=1}^{N}$ of the signal of interest x(t), acquired at the Nyquist rate (Equation 1) and expressed as the vector x, can be represented in terms of a basis of *N*-size orthonormal vectors ${\left\{\psi_{n}\right\}}_{n=1}^{N}$ and in terms of weighting coefficients ${\left\{\alpha_{n}\right\}}_{n=1}^{N}$:(2)$$x=\sum_{n=1}^{N}\psi_{n}\alpha_{n}=\Psi \alpha ,$$
where Ψ is the inverse Discrete Fourier Transform (DFT) matrix and α is the vector of the Fourier coefficients. In particular, if the vector x can be written as the combination of only *S* basis vectors, with S<<N, since only *S* of the coefficients ${\left\{\alpha_{n}\right\}}_{n=1}^{N}$ are non-zero, the vector α is *S*-sparse in the frequency domain. In this case, the inverse DFT matrix Ψ represents a sparsity basis. More generally, many vectors of acquired signals comply with the compressibility in the frequency domain, namely, they have many small coefficients but a few large coefficients in the frequency domain, where they can be concisely represented. Such condition is actually not rare: for instance, vectors of samples, related to signals such as audio, images or video, are compressible in the frequency domain by nature.

Sparsity is traditionally exploited by compression algorithms in order to lower digital data streams outgoing from ADCs. Indeed, in the *sample-then-compress* paradigm, firstly the signal is acquired at a fast sampling frequency and then part of the samples is discarded to reduce the size of the data. The paradox is that the desired compression relies on a later software-based step. Hence, the aforementioned compression algorithms reduce the data volume but they do not solve the issue related to the high sampling frequency. Likewise, the constraint on the size of the acquisition memory remains.

As long as fifteen years ago, the CS paradigm was proposed to overcome the limits imposed by the Nyquist-Shannon theorem (Equation 1) for wideband signals [5]. CS moves away from the sample-then-compress paradigm, since the input signal is directly acquired with fewer samples than those required at the Nyquist rate. Therefore, lowering sampling frequency and data rate of the output stream is the main advantage of the CS paradigm for wideband acquisition.

By way of illustration, Figure 1 shows the comparison between two generic data acquisition systems implementing the compression process. The former, based on the sample-then-compress paradigm, after the two blocks of sensing and conditioning, implements the traditional Analog/Digital (A/D) conversion. The digital signal requires to be compressed before being stored or transmitted; alternatively, if it has to be processed, the processing operates on uncompressed data. In the latter, based on the CS paradigm, the sub-Nyquist sampling substitutes for the traditional sampling. More simply, storage, transmission and processing are performed directly on a compressed (digital) signal.

## 3. Analog-to-Information Converters

An AIC is a device that, exploiting the CS paradigm, aims to acquire only the meaningful part of an analog signal, i.e., at the signal information rate. The terminology adopted to identify this architecture, although recalling the term ADC, properly wants to underline the ability to digitize not the whole input signal, but uniquely the related information content. The AIC working principle allows to merge in a single operation the two steps of acquisition and compression: an AIC compresses while sampling, and sampling frequency and data rate of the output stream are reduced at the same time. In AICs, signals are firstly acquired with fewer samples per second than needed according to the Nyquist-Shannon theorem, and then recovered through a reconstruction algorithm: the signal recovery is heavy, as the reconstruction algorithm is nonlinear. In other words, in AICs the front-end is lightened by a considerable burden that is relegated to the back-end. The functioning of such an innovative technology is particularly efficient also for other reasons. In fact, a lower data rate, for a given information content, implies a reduction of occupancy of both acquisition memory and bandwidth used for transmission, suggesting the use of AICs for Internet-of-Things and in support of 5G. Moreover, compared to ADCs, the overall energy needed for the power supply section is lowered by reducing the number of samples per second [6,7]. On the other hand, AICs present front-ends that include signal generators, mixers and filters. In recent prototypes, able to perform high compression and ultra-wideband spectrum sensing [8,9], the architectural complexity increases further [7,8,9]. The main trade-off of the greater quantity of hardware components in AICs is the higher level of distortion and noise. Besides, due to the huge number of involved variables whose the reconstructed signal is function, the difficulty to evaluate the uncertainty in AICs should be considered. Compared to the ADC output, the input signal reconstruction in AICs is more sensitive to hardware non-idealities, such as aperture and jitter effects [10]. Hardware non-idealities can be identified by characterizing the AIC front-end and its single building blocks. However, the architectural complexity implicates an intricate process for front-end characterization [3,6]. Table 1 summarizes the advantages and the limitations of AIC-based measurement instrumentation in comparison to traditional measurement instrumentation employing ADCs.

Although AICs are not an industrial reality yet, they are a promising perspective of research within different fields. The prototypes developed in the last few years prove the AIC potential from building structural monitoring [11] to remote healthcare systems [12,13,14], from electrical impedance spectroscopy [15] to spectrum sensing [6,7,8,9,16,17,18,19,20,21,22,23,24,25]. Indeed, as proposed in [11], the AIC is exploited to acquire acoustic emission signals. In smart living environments, AICs can provide a valid support as measurement instrumentation for biomedical signals [12]. For instance, as shown by the experimental results of the prototype [13], electrocardiogram (ECG) signals are reconstructed starting from a signal sub-sampled at a frequency 8 times smaller than the Nyquist rate. Furthermore, in [13], a numerical evaluation of the proposed AIC is shown for electroencephalogram (EEG) signals. Instead, the prototype [14], in Complementary Metal–Oxide–Semiconductor (CMOS) technology, is designed to monitor with low power consumption different types of biomedical signals, i.e., ECG, EEG, photoplethysmogram, electromyogram and electrocorticogram. The operation mode of the prototype [14], experimentally demonstrated for ECG signals, highlights that, despite the compression phase, the signal information content, such as QRS complex, is preserved. Anyway, since AIC technology allows to reduce high sampling frequencies as well as large amounts of samples, the most investigated application is spectrum sensing, where AICs can be employed as code division multiple access receivers [16] or wideband receivers [6,7,8,9,17,18,19,20,21,22,23,24,25]. Some wideband receivers perform the function of radar detection [22,23,24] or cognitive radio [7,9]. AICs for wideband spectrum sensing are implemented as both printed circuit board assemblies [17,18,19,21,22] or integrated circuits [6,7,8,9,23,24,25]. The chip technology can be CMOS [6,7,24], Bipolar CMOS [8], Indium Phosphide Heterojunction Bipolar Transistor [23,25], or the chip can be an Application Specific Integrated Circuit [9]. These AICs implemented in integrated circuits sub-sample at very low sampling frequencies, in comparison to the large input bandwidths they can reach. Just to mention an example, one of the most recent chip implementations, proposed in [8], has a broad input bandwidth of 2 GHz to 18 GHz and it can sub-sample at a frequency 128.5 times smaller than the Nyquist rate. Similarly, the prototype proposed in [9] has an input bandwidth of 0 GHz to 16 GHz and it sub-samples at a frequency of only 125 MHz.

### Acquisition and Reconstruction

AICs consist of both hardware and software components and are typically characterized by two main sections: an acquisition front-end and a reconstruction back-end. The generic framework of an AIC is depicted in Figure 2.

The acquisition front-end is devoted to sub-sampling the input signal and is composed of a conditioning block and one or more ADCs. The input signal x(t) enters the device and *M* samples are acquired by the front-end, which implements the sampling below the Nyquist frequency. For a given acquisition time window, the number *M* of the samples acquired by the AIC front-end is reduced in comparison to the number *N* of the samples that would be acquired at the Nyquist rate. Therefore, the acquisition process of the AIC front-end is a compression of the Nyquist rate acquisition. Given the vector x of the *N* samples acquired at the Nyquist rate, the vector of the *M* samples acquired by the AIC is ideally modeled as a linear transformation of x [1,2]:(3)$$y_{mod}=\Phi x,$$
where Φ is an M×N*measurement matrix* representing the compression process of the AIC according to the compression ratio K =N/M. It is important to highlight that Equation (Equation 3) emulates the physical path of the signal inside the architecture and provides an ideal output. The acquisition front-end, for its part, is a real system and does not behave ideally. In other words, differently from what happens in applications that are only software-based, the real output vector y, that is acquired by the AIC front-end, is not a linear transformation exactly identified through the ideal measurement matrix Φ, but it is a more generic function of x:(4)y=g(x),
where g(·) can even be non-linear in practice and includes potential non-idealities of the architecture, conversely to what is supposed in (Equation 3).

The reconstruction back-end is devoted to estimating the input signal and is composed of a processing unit. Starting from the output signal y, the back-end recovers the input signal via a minimization algorithm that searches for the sparsest solution. By expressing the vector x in terms of the sparsity basis Ψ and the coefficient vector α, the maximally sparse solution can be found by looking for an estimate of α as follows [26]:(5)$$\hat{\alpha }=\min_{\alpha }{∥\alpha ∥}_{1},\;\;\;\;\mathrm{subject}\;\mathrm{to}:\;\Phi \Psi \alpha =y.$$
Lastly, the solution to problem (Equation 5) can be brought in the time domain: $\hat{x}=\Psi \hat{\alpha }$. It must be noted that, to find the sparsest solution, an $l_{0}$ quasi-norm counting the number of non-zero elements should be adopted instead of the $l_{1}$ norm. Unfortunately, the minimization by means of the $l_{0}$ norm is computationally prohibitive, even for vectors of modest size. Anyway, in [5,26], the solution to the minimization of the $l_{0}$ norm is proven to be equivalent to the minimization of the $l_{1}$ norm.

In order to estimate the input signal through the $l_{1}$ norm, another essential condition must be fulfilled in the reconstruction step in addition to sparsity. Such condition is related to the degree of similarity between the measurement matrix Φ and the sparsity basis Ψ, defined as mutual coherence [26]:(6)$$\mu \left(\Phi ,\Psi \right)=\max\left\{|<\phi_{i},\psi_{j}>|:\phi_{i}\in \Phi ,\psi_{j}\in \Psi \right\},$$
where $\phi_{i}$ are row vectors, with i=1,2,⋯,M, and $\psi_{j}$ are column vectors, with j=1,2,⋯,N. The parameter μ roughly indicates how much the measurement matrix Φ is alike to the sparsity basis Ψ: if the row vectors $\phi_{i}$ and the column vectors $\psi_{j}$ are correlated, the coherence is large, otherwise it is small. The necessary and sufficient condition for the minimization algorithm to recover correctly an *S*-sparse signal is the *incoherence property*. The incoherence property states that the number of the measurements must comply with the following inequality [27]:(7)$$M\geq C\mu^{2}\left(\Phi ,\Psi \right)S\log N,$$
where *C* is a generic small positive constant. In other words, any input signal may be reconstructed from a set of measurements of a minimum number that is $O(\mu^{2}S\log N)$, and, the lower the coherence is, the lower the required number *M* is. The incoherence property can be easily satisfied, for example, if the measurement matrix is chosen with random entries from a Bernoulli or Gaussian distribution [5,26,28,29], but it can also be satisfied when the measurement matrix is properly built with deterministic entries [30,31,32].

## 4. Metrological Characterization in AICs

As shown in previous Section 3, according to (Equation 5), both the actual samples of the vector y and the ideal elements of the measurement matrix Φ are essential to estimate the input signal. In detail, the measurement matrix introduced by (Equation 3) is a purely mathematical model that, conceived in the design phase of the AIC prototype, tries to reproduce the behaviour of the architecture [18,33]. Instead, hardware components of the front-end are characterized by several non-idealities not contemplated by the measurement matrix in (Equation 3). The choice of the measurement matrix in the minimization problem (Equation 5) affects the accuracy in the signal recovery. Generally, the performance of AICs conditions the measurement instrumentation based on them, just like the performance of ADCs conditions the traditional measurement instrumentation. Obviously, the need for a more or less accurate reconstruction depends on the specific field of application where the AIC-based measurement instrumentation is employed. For example, the features of a signal measured by a vector signal analyzer should be more accurate than for radar detection.

In order to properly recover the input signal, a realistic model of the acquisition front-end should be designed such that it can better represent the vector of the actual measurements (Equation 4). Specifically, two different directions could be chosen to obtain a model closer to the physical reality of the device. The former direction consists in considering the circuit models of the components employed in the prototype. Nevertheless, such direction often turns out to be impractical because, in the vast majority of cases, the designer (or the manufacturer) does not provide the circuit models of the front-end components. The latter direction—frequently the only possible one—consists in black-box modeling. For the purpose of black-box modeling, several investigations should be carried out to attain a measurement matrix indicative of the AIC architecture: (i) definition of the numerical model and development of behavioural simulations for model assessment; (ii) validation of the model on the prototype and metrological characterization of the prototype; (iii) identification of hardware non-idealities and correction of the model. In other words, the problem of correct modeling should be addressed through metrological characterization. AIC characterization manages to improve the accuracy of the measurement matrix in representing the architecture. As a consequence, characterization plays a fundamental role in AICs [3], even more important than in ADCs. Indeed, while metrologically characterizing ADCs is intended to test the digitized version of the input signal (the ADC output vector) [34], metrologically characterizing AICs is instead functional to test and obtain correctly the digitized version of the input signal (the reconstructed vector $\hat{x}$). In particular, identification of front-end imperfections by means of characterization is of crucial relevance and the different non-idealities should be included in the mathematical model.

Quantifying systematic effects and correcting the model accordingly do not remove random phenomena such as random jitter, supply voltage instability, temperature variations, thermal noise, imperfect synchronization, channel cross-talk. For this reason, the issue of hardware non-idealities and estimate robustness is typically faced by involving an error term in the reconstruction step. The measurement procedure is considered as affected by additive noise e, i.e., a vector of deterministic or random errors ${\left\{e_{m}\right\}}_{m=1}^{M}$ whose norm obeys ${∥e∥}_{2}\leq \varepsilon$, with ε a small positive constant representing a threshold. The reconstruction problem (Equation 5) is relaxed to [35]:(8)$$\hat{\alpha }=\min_{\alpha }{∥\alpha ∥}_{1},\;\;\;\;\mathrm{subject}\;\mathrm{to}:\;{∥\Phi \Psi \alpha -y∥}_{2}\leq \varepsilon .$$
However, it should be pointed out that the additive noise term e generally contemplates simple and uncorrelated errors in the sub-sampled data [36,37] and cannot cover up all types of non-idealities. Thus, including the noise term in the reconstruction problem is not enough, and modeling the architecture correctly through metrological characterization is an important procedure for obtaining a signal reconstruction that is as accurate as possible.

### 4.1. Characterization Procedure for AICs

Metrologically characterizing AICs is an intricate task because of the architectural complexity and, thus, the joint effect of several hardware non-idealities. As a matter of fact, distortion, internal noise and other architectural non-idealities intervene simultaneously and reduce the AIC performance overall. Currently, official procedures for AIC characterization have not been approved. To this aim, the standardization of characterization procedures in AICs could be very useful. In reality, standardization activities of characterization procedures are not operative hitherto, since AICs are not commercial devices yet. Anyway, a standardization initiative can be undertaken within the “Technical Committee 10—Waveform Generation, Measurement & Analysis” of the IEEE Instrumentation and Measurement Society [38].

Several AIC architectures have been proposed in the scientific literature so far. Each architecture stands out for a given acquisition mechanism. For various applications, in the greatest part of the implemented prototypes, the AIC architectures make use of a modulation pre-sampling block. This pre-sampling block preserves the information content by modulating/demodulating the input signal through a Pseudo Random Binary Sequence (PRBS), as in the Random Demodulator (RD) [39] and the Modulated Wideband Converter (MWC) [21], or through a Wavelet signal, as in the Non-Uniform Wavelet Bandpass Sampler (NUWBS) [9]. In some architectures, AICs are designed on parallel channels. The parallel channels can either contain an ADC each [6,7,8,21,22,23,24] or they can all converge in an adder circuit followed by a single ADC [40,41]. Metrological characterization for the analysis of non-idealities should be expanded in such AIC architectures by evaluating whether the parallel channels need to compensate for similar building blocks (especially ADCs) having different imperfections, such as skew of the sampling clock, but also gain, offset and phase mismatch [42] as it occurs in time-interleaved ADCs.

Due to architectural AIC variety, the metrological characterization should be addressed in several steps based on architecture-specific experiments [3]. In the following, a characterization procedure is described. Characterization requires, firstly, that the building blocks of the AIC prototype be fully analyzed and considered from a system-level perspective. To this end, investigating the state-of-the-art AIC architectures, prototypes and models can be useful. Secondly, the potential non-idealities occurring in AICs should be deeply investigated. In this respect, the non-idealities affecting AICs should be differentiated in deviations from the model, distortion and noise, by ascribing distortion and noise to the building blocks from which they originate. In the following step, testing methods should be developed. AICs can be tested essentially by means of two typologies of methods. In the first typology, the AIC is tested by observing the sub-sampled vector y [18,21,43,44,45]. In the second typology, the AIC is instead tested by observing the reconstructed vector $\hat{x}$ [6,7,13,19,20,23,24,25,39]. Figures of merit are identified on the basis of the typology of the chosen testing method. Then, sine waves are given as input to the AIC depending on increasing frequency values, and the figures of merit are evaluated through the vector y or the vector $\hat{x}$. Suitable figures of merit from the ADC literature, such as Signal-to-Noise Ratio [19,39] or Effective Number of Bits [6,25], can be contextualized to assess the performance in AICs. Otherwise, novel figures of merit can be adopted too, such as the Model Error (ME) proposed for performance assessment before the reconstruction [43,44,45]:(9)$$\mathrm{ME}=20\log\left\{\frac{\sqrt{\sum_{m=1}^{M}{\left[y\left(m\right)-y_{mod}\left(m\right)\right]}^{2}}}{\sqrt{\sum_{m=1}^{M}y^{2}\left(m\right)}}\right\},$$
where ${\left\{y\left(m\right)\right\}}_{m=1}^{M}$ and ${\left\{y_{mod}\left(m\right)\right\}}_{m=1}^{M}$ are the sets of the *M* samples, respectively, of the real output vector y (Equation 4) and the vector $y_{mod}$ (Equation 3) obtained according to a certain mathematical model. Finally, the characterization can be implemented by assessing the AIC performance depending on the non-idealities, in terms of the figures of merit.

AIC characterization can be achieved by considering either the complete acquisition chain or its components separately. In the former case, the measurement matrix Φ is calibrated to include the hardware non-idealities of the whole acquisition chain in the AIC model. In such calibration, the AIC prototype is characterized in response to different input sinewaves [6,39,44,45]. Due to long times, however, building a calibration model for the measurement matrix can become impractical. In the latter case, instead, the single building blocks are characterized one-by-one in response to different input sinewaves [44,46,47]. The measurement matrix Φ is subsequently built starting from the calibration models related to the single building blocks [44,46].

### 4.2. Experimental Testing of AICs

In order to prove the efficacy of the characterization approach illustrated above, the AIC architectures based on the modulation pre-sampling block have been experimentally tested by mastering actual prototypes. A perspective is illustrated in Table 2. Table 2 details: (i) the working principle, i.e., the carrier signal that modulates the input signal, the type of sampling (uniform or non-uniform), the mathematical model (in time or frequency domain); (ii) the original prototype; (iii) the research works concerning experimental testing in AICs. Generally speaking, if on one side characterizing AICs underlines the limitations in models due to non-idealities, on the other side it can suggest architectural changes to reduce the effects of such non-idealities. In particular, the strong points of AIC characterization are discussed below.

Metrological characterization allows, first of all, to ponder the ability and the limitation of linear modeling in representing an AIC prototype. In this regard, two different linear models for the same architecture are considered in [44]. In detail, the experimentally acquired output signal is compared to the output signals obtained according to both a time domain model and a frequency domain model. The time domain model is more able to represent the prototype. As evidence, Figure 3 shows the ME (averaged on 100 runs) versus the frequency of the input signal for the two models. As illustrated by Figure 3, for the time domain model, the ME obtained with the compression ratio K =4 almost overlaps the ME obtained with the compression ratio K =1 (namely when compression is not performed). Instead, for the frequency domain model, the ME is higher and it is also heavily affected by the compression ratio K. The main reason behind a similar result is that the parameters of the time domain model, which are determined by a characterization phase, are inclusive of non-idealities along the full path of the signal inside the architecture. Conversely, in the model in the frequency domain, where only the low-pass filter is characterized, other component non-idealities are not included.

Characterizing the building blocks helps to draw attention to non-idealities in the whole chain from the input to the output of the AIC. Let the RD be considered. In such architecture, the jitter of each PRBS symbol is one of the prevalent non-idealities due to the symbol rate, higher than the ADC sampling frequency. Jitter consists of a stochastic component, the Random Jitter, and a systematic component, the Deterministic Jitter [48]. Although only the Random Jitter effect is typically evaluated in AICs [6,10], the Deterministic Jitter also has a significant weight, especially the sub-component of Duty Cycle Distortion. In fact, as proven in [43], by characterizing the PRBS waveform generated by the arbitrary waveform generator employed in the RD, the weight of Duty Cycle Distortion (216.0 ps) results considerably greater than the weight of Random Jitter (5.4 ps). In [43], a sensitivity analysis is carried out to quantify the ME resulting from approximating in AIC the PRBS as non-affected by jitter. Figure 4 presents the ME depending on the weight of Duty Cycle Distortion and Random Jitter. The values start from the condition of the absence of jitter and, then, they progressively increase. Worth noting is that, if the weights of the jitter components were equal, Random Jitter had more incidence than Duty Cycle Distortion. However, as Duty Cycle Distortion weighs more, both the components should be taken into account in the model of the architecture. Furthermore, in some cases, similar non-idealities can be noticed in various architectures, such as non-idealities belonging to the blocks of signal generators, filters and ADCs. Specifically, quantization noise is a non-ideality common to all AIC architectures, as related to ADCs. It too should be taken into account in the model, since it degrades the reconstruction performance significantly when the number *S* of sparse components increases [20].

Lastly, a key aspect is that characterization in AICs can suggest changing architectural components on the basis of the analysis of non-idealities. For instance, the wideband operation of the mixer, employed in AICs for the modulation of the input signal, introduces a high distortion component due to intermodulation [47]. Nonetheless, as proven by the results of [45], here reported in Figure 5, the analog multiplier can be employed as an alternative because it reveals experimentally a behavior closer to the ideal multiplication, helping the prototype to reduce the distortion effect. As a matter of fact, the prototype based on the analog multiplier, compared to the time domain model, exhibits a lower ME (averaged on 20 runs) than the prototype based on the mixer. On the other hand, characterization can also demonstrate that changing a specific component in the architecture does not alter the reconstruction performance. As an example, if the NUWBS employs a low-pass filter before the ADC, it reaches a performance that is comparable to that of the original architecture employing an analog integrator [20].

## 5. Conclusions

In this article, a perspective on AICs has been provided in terms of advantages and technical issues related to their use in measurement instrumentation. Specifically, the two AIC sections of acquisition and reconstruction have been described. The advantages resulting from such devices are numerous for wideband acquisition, since the sampling frequency is reduced. The samples are not acquired in large quantities and then compressed, but rather in reduced quantities. The functioning of AICs is particularly efficient, suggesting their use for Internet-of-Things and in support of 5G.

The signal reconstruction in AICs influences the measurement instrument based on them: an accurate modeling of the architecture is required to properly reconstruct the input signal. Metrological characterization of AIC building blocks play a fundamental role in this sense. Indeed, characterizing AICs can point out the limitation of linear modeling, due to non-idealities related to given building blocks, but it can also suggest architectural changes to reduce the effects of such non-idealities.

An important note is that research in the field of AIC-based measurement instrumentation is in progress. A good part of the architectures proposed in the literature do not correspond hitherto to prototypes actually implemented on printed circuit board or chip. Therefore, AIC characterization and testing are still open problems.

## Figures and Tables

**Figure 1 sensors-23-00861-f001:**
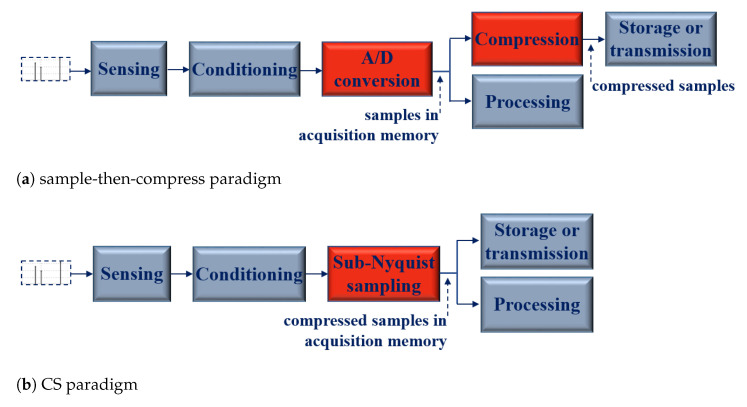
Comparison between the data acquisition systems implementing compression: (**a**) sample-then-compress paradigm and (**b**) CS paradigm.

**Figure 2 sensors-23-00861-f002:**
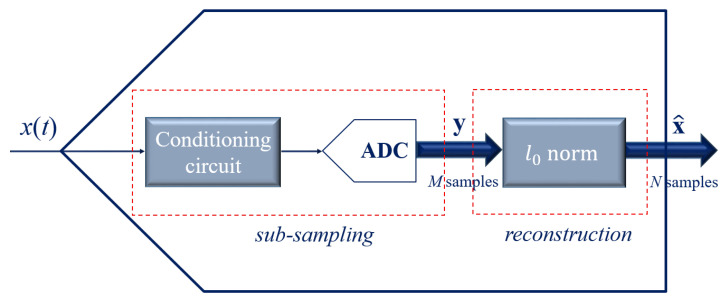
Generic framework of an AIC with its two sections: the acquisition front-end and the reconstruction back-end.

**Figure 3 sensors-23-00861-f003:**
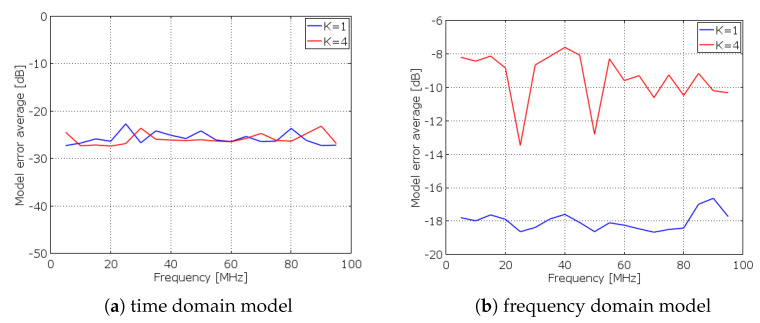
Experimental results of ME in the RD for (**a**) the time domain model and (**b**) the frequency domain model [44].

**Figure 4 sensors-23-00861-f004:**
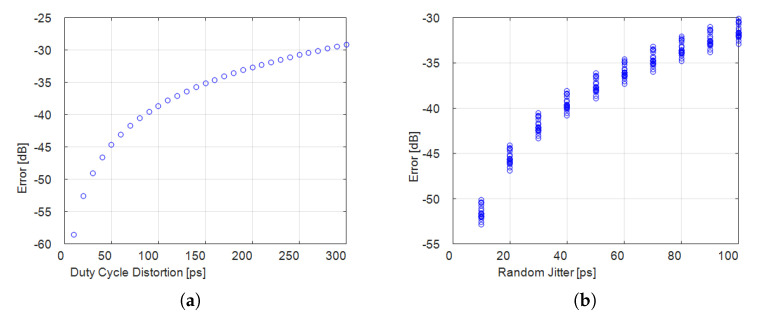
Numerical results of ME in the RD versus (**a**) Duty Cycle Distortion and (**b**) Random Jitter [43].

**Figure 5 sensors-23-00861-f005:**
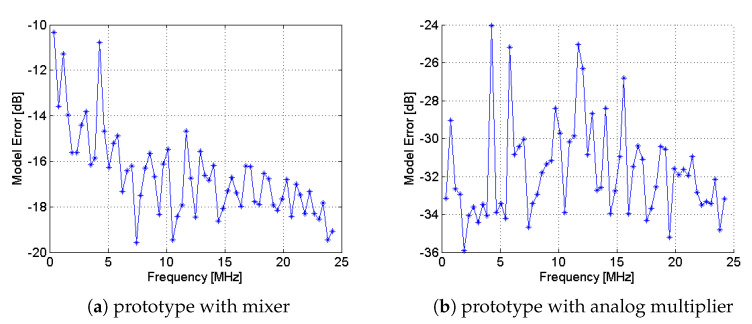
Experimental results of ME in the RD based on (**a**) the mixer and (**b**) the analog multiplier [45].

**Table 1 sensors-23-00861-t001:** AIC-based measurement instrumentation: advantages and limitations.

Advantages	Limitations
Lower sampling frequency	Sparsity requirement on input signal
Lower data rate	More computational load for back-end
Reduced occupancy of acquisition memory	Architectural complexity [7,8,9]
Reduced power consumption [6,7]	Difficult uncertainty evaluation
High compression in recent architectures [8,9]	More sensitivity to non-idealities [10]
Broadband input in recent architectures [8,9]	Intricate front-end characterization [3,6]

**Table 2 sensors-23-00861-t002:** AIC architectures: working principle, original prototype and testing.

Architecture	Carrier	Sampling	Model	Original Prototype	Testing
RD	PRBS	uniform	time domain	[39]	[44,45]
MWC	PRBS	uniform	frequency domain	[21]	[44]
NUWBS	Wavelet	non-uniform	time domain	[9]	[20]

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
