# Peer review of "Over the Limits of Traditional Sampling: Advantages and Issues of AICs for Measurement Instrumentation"

_sensors, 2023, doi:10.3390/s23020861_

Round 1

Reviewer 1 Report

Dear Authors,

The layout of the article is well organized. The article is well written.

I have no major comments on the content of the article.

Best regards

Author Response

The authors thank the Reviewer for the appreciation of the work.

Reviewer 2 Report

Comments,

   Analog-to-Information Converters (AICs) represent the implementation of the Compressed Sensing theory in practice.

   This paper introduced an Analog-to-Information Converter (AIC) method aimed at acquiring efficiently wideband signals, and discussed employing AICs in measurement instrumentation and the analysis of their building blocks and the corresponding non-idealities in order to improve the signal reconstruction. So it has referential value in a certain sense. However, the following questions should be considered and explained before the paper can be further considered for publishing.  

(1)The method proposed in the paper, is it feasible to design multi-channel architecture  AICs? How to test and calibrate AICs?

2)AIC characterization and standardization issues need to be explained.

This is expectant that the manuscript would be further improved and explained.

Author Response

The authors thank the Reviewer for the useful suggestions that improved the scientific content and readability of the manuscript.

In the following, the remarks of the Reviewer are reported in black, the corresponding authors’ comments in blue, the added text in red.

Comments,

   Analog-to-Information Converters (AICs) represent the implementation of the Compressed Sensing theory in practice.

   This paper introduced an Analog-to-Information Converter (AIC) method aimed at acquiring efficiently wideband signals, and discussed employing AICs in measurement instrumentation and the analysis of their building blocks and the corresponding non-idealities in order to improve the signal reconstruction. So it has referential value in a certain sense.

The authors thank the Reviewer for the appreciation of the work.

However, the following questions should be considered and explained before the paper can be further considered for publishing. 

(1)The method proposed in the paper, is it feasible to design multi-channel architecture  AICs? How to test and calibrate AICs?

By following the Reviewer suggestion, the issue related to multi-channel architectures was added. The modified text is reported below:

In some architectures, AICs are designed on parallel channels. The parallel channels can either contain an ADC each [6-8,21-24] or they can all converge in an adder circuit followed by single ADC [37,38]. Metrological characterization for analysis of non-idealities should be expanded in such AIC architectures by evaluating whether the parallel channels need to compensate for similar building blocks (especially ADCs) having different imperfections, such as skew of sampling clock, but also gain, offset and phase mismatch [39], as it occurs in time-interleaved ADCs.

The modified text related to AIC testing is instead reported below:

AICs can be tested essentially by means of two typologies of methods. In the first typology, the AIC is tested by observing the sub-sampled vector y [18,21,40-42]. In the second typology, the AIC is instead tested by observing the reconstructed vector x [6,7,13,19,20,23-25,36]. Figures of merit are identified on the basis of the typology of the chosen testing method. Then, sine waves are given as input to the AIC depending on increasing frequency values, and the figures of merit are evaluated through the vector y or the vector x.

Finally, the modified text related to AIC calibration is reported below:

AIC characterization can be achieved by considering either the complete acquisition chain or its components separately. In the former case, the measurement matrix Φ is calibrated to include the hardware non-idealities of the whole acquisition chain in the AIC model. In such calibration, the AIC prototype is characterized in response to different input sinewaves [6,36,41,42]. Due to long times, however, building a calibration model for the measurement matrix can become impractical. In the latter case, instead, the single building blocks are characterized one-by-one in response to different input sinewaves [41,43,44]. The measurement matrix Φ is subsequently built starting from the calibration models related to the single building blocks [41,43].

(2)AIC characterization and standardization issues need to be explained.

This is expectant that the manuscript would be further improved and explained.

The authors thank the Reviewer for this comment. The AIC characterization and standardization issues were explained. Moreover, the Section 4 was re-organized more precisely. In fact, it now includes the Subsection 4.1 “Characterization procedure for AICs”. The modified text is reported below:

Metrologically characterizing AICs is an intricate task, because of the architectural complexity and, thus, the joint effect of several hardware non-idealities. As a matter of fact, distortion, internal noise and other architectural non-idealities intervene simultaneously and reduce the AIC performance overall. Currently, official procedures for AIC characterization have not been approved. To this aim, the standardization of characterization procedures in AICs could be very useful. In reality, standardization activities of characterization procedures are not operative hitherto, since AICs are not commercial devices yet. Anyway, a standardization initiative can be undertaken within the "Technical Committee 10 - Waveform Generation, Measurement & Analysis" of the IEEE Instrumentation and Measurement Society [35].

Reviewer 3 Report

The content of this paper is to introduce an architecture to acquire its efficiently wideband signals, named as the Analog-to-Information Converter (AIC). The working principle of the sub-Nyquist sampling is also proposed.

This paper provided some useful information about the AIC. It corresponded with the requirement of the “Prospective” scope. However, the content of this paper could be enhanced.

1.The advantages of AIC are described. However, AICs are not an industrial reality yet. Please lists its disadvantages and limitation (including the literature) and describes the way to improve.

2.A table to list the recent literature about this AIC technique including the advantages and disadvantages of AIC will be very helpful for readers.

3. The authors listed some literature about the application of AIC (lines 121-124, literature 6-9). Please provide more information about this research. Besides this literature (6-9), please supply more literature about the application of AIC.

Author Response

The authors thank the Reviewer for the useful suggestions that improved the scientific content and readability of the manuscript.

In the following, the remarks of the Reviewer are reported in black, the corresponding authors’ comments in blue, the added text in red.

The content of this paper is to introduce an architecture to acquire its efficiently wideband signals, named as the Analog-to-Information Converter (AIC). The working principle of the sub-Nyquist sampling is also proposed.

This paper provided some useful information about the AIC. It corresponded with the requirement of the “Prospective” scope.

The authors thank the Reviewer for the appreciation of the work.

However, the content of this paper could be enhanced.

1.The advantages of AIC are described. However, AICs are not an industrial reality yet. Please lists its disadvantages and limitation (including the literature) and describes the way to improve.

By following the Reviewer suggestion, AIC disadvantages and limitation (including literature) are listed together with the way to improve them. The modified text is reported below:

On the other hand, AICs present front-ends that include signal generators, mixer and filters. In recent prototypes, able to perform high compression and ultra-wideband spectrum sensing [8,9], the architectural complexity increases further [7-9]. Anyway, the main trade off in AICs is the level of distortion and noise, due to a greater number of hardware components. Besides, it should be considered that the difficulty to evaluate the uncertainty in AICs is high, due to the huge number of involved variables whose the reconstructed signal is function. Compared to ADC output, the input signal reconstruction in AICs is more sensitive to hardware non-idealities, as aperture and jitter effects [10]. Hardware non-idealities can be identified by characterizing the AIC front-end and its single building blocks. However, the architectural complexity implicates an intricate process for front-end characterization [3,6].

2.A table to list the recent literature about this AIC technique including the advantages and disadvantages of AIC will be very helpful for readers.

As required by the reviewer, a Table with the recent literature was added to the paper: Table 1. AIC-based measurement instrumentation: advantages and limitations.

3. The authors listed some literature about the application of AIC (lines 121-124, literature 6-9). Please provide more information about this research. Besides this literature (6-9), please supply more literature about the application of AIC.

The authors thank the Reviewer for this comment. More information about literature [6-9], now corresponding to [11-14] was provided. Besides literature [11-14] was provided more literature about the application of AICs. The modified text is reported below:

The prototypes developed in last few years witness the AIC potential from building structural monitoring  [11] to remote healthcare systems [12-14], from electrical impedance spectroscopy [15] to spectrum sensing [6-9,16-25]. Indeed, as proposed in [11], the AIC is exploited to acquire acoustic emission signals. In smart living environments AICs can provide a valid support as measurement instrumentation for biomedical signals [12]. For instance, as shown by the experimental results of the prototype [13], electrocardiogram (ECG) signals are reconstructed starting from a signal sub-sampled at a frequency 8 times smaller than the Nyquist rate. Moreover, in [13] a numerical evaluation of the proposed AIC is shown for electroencephalogram (EEG) signals. Instead, the prototype [14], in Complementary Metal–Oxide–Semiconductor (CMOS) technology, is designed to monitor different types of biomedical signals, i.e. ECG, EEG, photoplethysmogram, electromyogram and electrocorticogram, with low power consumption. The operation mode of the prototype [14], experimentally demonstrated for ECG signals, highlights that, despite the compression phase, the signal information content, such as QRS complex, is preserved. In any case, since AIC technology allows to reduce high sampling frequencies as well as large amounts of samples, the most investigated application is spectrum sensing, where AICs can be employed as code division multiple access receivers [16] or wideband receivers [6-9,17-25]. Some wideband receivers perform the function of radar detection [22-24] or cognitive radio [7,9]. AICs for wideband spectrum sensing are implemented as both printed circuit board assemblies [17-19,21,22] or integrated circuits [6-9,23-25]. The chip technology can be CMOS [6,7,24], Bipolar CMOS [8], Indium Phosphide Heterojunction Bipolar Transistor [23,25], or the chip can be an Application Specific Integrated Circuit [9]. These AICs implemented in integrated circuits sub-sample at really low sampling frequencies, in comparison to the large input bandwidths they can reach. Just to mention an example, one of the most recent chip implementation, proposed in [8], has a broad input bandwidth of 2 GHz to 18 GHz and it can sub-sample at a frequency 128.5 times smaller than the Nyquist rate. Similarly, the prototype proposed in [9], has an input bandwidth of 0 GHz to 16 GHz and it sub-samples at a frequency of only 125 MHz.

Round 2

Reviewer 3 Report

The content of the revised manuscript has been improved significantly.